# Single-cell modeling of routine clinical blood tests reveals transient dynamics of human response to blood loss

Anwesha Chaudhury[1,2]\*, Geoff D Miller[3], Daniel Eichner[3], John M Higgins[1,2]\*

[1]Center for Systems Biology and Department of Pathology, Massachusetts General Hospital, Boston, United States; [2]Department of Systems Biology, Harvard Medical School, Boston, United States; [3]Sports Medicine Research and Testing Laboratory, Salt Lake City, United States

**Abstract** Low blood count is a fundamental disease state and is often an early sign of illnesses including infection, cancer, and malnutrition, but our understanding of the homeostatic response to blood loss is limited, in part by coarse interpretation of blood measurements. Many common clinical blood tests actually include thousands of single-cell measurements. We present an approach for modeling the unsteady-state population dynamics of the human response to controlled blood loss using these clinical measurements of single-red blood cell (RBC) volume and hemoglobin. We find that the response entails (1) increased production of new RBCs earlier than is currently detectable clinically and (2) a previously unrecognized decreased RBC turnover. Both component responses offset the loss of blood. The model provides a personalized dimensionless ratio that quantifies the balance between increased production and delayed clearance for each individual and may enable earlier detection of both blood loss and the response it elicits.

**\*For correspondence:**
achaudhury@mgh.harvard.edu
(AC);
john_higgins@hms.harvard.edu
(JMH)

**Competing interests:** The authors declare that no competing interests exist.

## Introduction

Single-cell measurements and models promise to capture important biological heterogeneity and reveal novel mechanisms (*Baron et al., 2018*; *Giustacchini et al., 2017*; *Shalek et al., 2013*; *Tusi et al., 2018*). Routine clinical blood tests already include single-cell measurements of cellular, nuclear, and cytoplasmic morphology and some single-cell protein concentrations (*Chaudhury et al., 2017*; *Higgins and Mahadevan, 2010*; *Kim and Ornstein, 1983*; *Mohandas et al., 1986*). These clinical assays measure fewer states per cell (~1–10) than more recently developed single-cell molecular methods (>1000) (*Shalek et al., 2013*; *Tusi et al., 2018*), but these clinical data have three strengths for modeling: (1) the low-dimensional state space is densely sampled, (2) existing mechanistic understanding of single-cell trajectories in this state space can guide specification of dynamic equations, and (3) there is a shorter path to clinical translation of any potential insights. The typical adult human produces about 2 million RBCs per second, with a similar rate of clearance of old RBCs after they have circulated for ~90–120 days. RBC lifespan is tightly controlled within each person but varies from one person to the next (*Cohen et al., 2008*; *Malka et al., 2014*). The volume of a typical RBC decreases by about 30% and the hemoglobin mass by about 20% over the course of the RBC's lifespan, with the average hemoglobin concentration ([Hb]) increasing modestly (*Malka et al., 2014*; *Willekens et al., 2008*). Routine complete blood counts (CBCs) can include measurements of single-cell volume ($v$) and hemoglobin ($h$) for ~50,000 individual RBCs (*Figure 1*). Some of the youngest RBCs ('reticulocytes' <~3 days old) can be identified in these counts because they generally have RNA remnants in their membranes (*d'Onofrio et al., 1995*). The typical healthy RBC follows a ($v,h$)-trajectory along the major axis of the ($v,h$) distribution ($u$ in *Figure 1*) as it ages until eventually being cleared in the lower

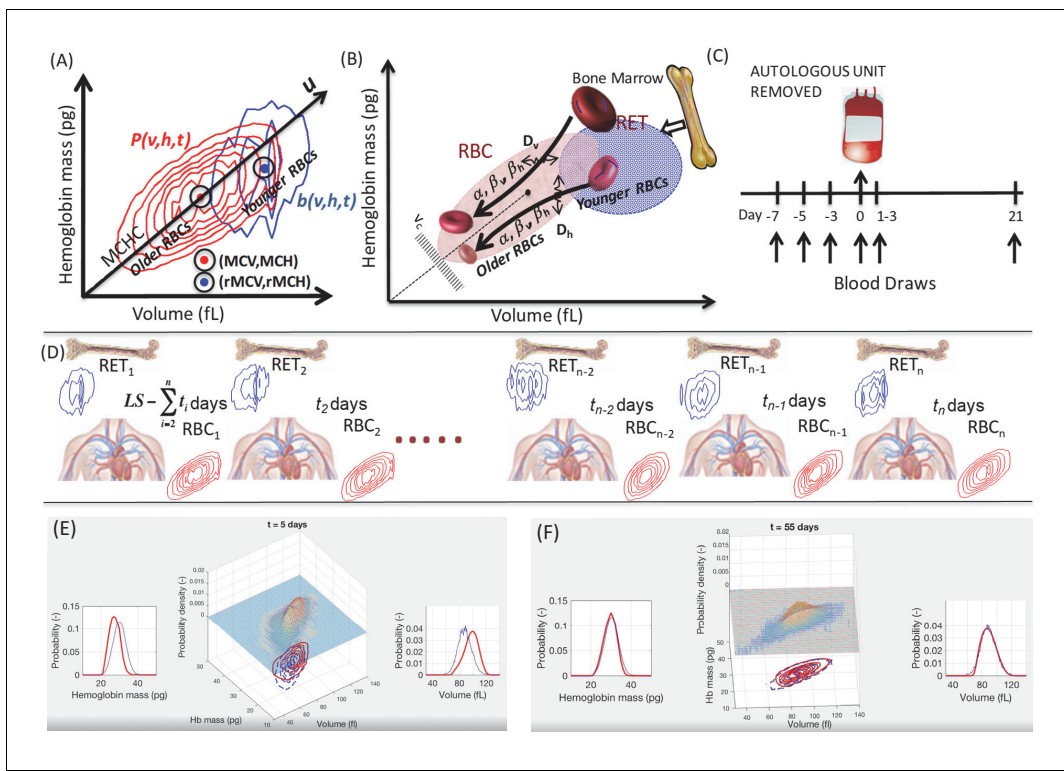

**Figure 1.** Unsteady-state modeling of single-RBC volume and hemoglobin dynamics. (**A**) Routine complete blood counts (CBC) measure the single-cell volume and hemoglobin for young RBCs ($b(v,h,t)$, blue contours showing 'reticulocytes' that are $<{\sim}3$ days old) as well as for all circulating RBCs ($P(v,h,t)$, red contours showing RBCs of all ages from 0 up to 90–120 days, with RBC lifespan well-controlled in each person but varying from one person to the next). The black line through the origin shows the mean hemoglobin concentration (mean corpuscular hemoglobin concentration, MCHC) for the sampled population and this major axis of the distribution (**u**) provides a very rough estimate of RBC age, with higher $u$ corresponding to younger age. (**B**) Schematic of the model of single-RBC volume-hemoglobin dynamics. Individual RBCs are produced as reticulocytes (RET) in the top right and lose about 30% of their volume and about 20% of their hemoglobin during their 90–120 day lifespan, with volume and hemoglobin reductions occurring during an early fast phase parameterized by $\beta_v$ and $\beta_h$ and a later slow phase parameterized by $\alpha$, with fluctuations in rates of single-RBC volume and hemoglobin change quantified by $D_v$ and $D_h$. As the single-RBC volume and hemoglobin continue to fall, the probability of clearance increases dramatically as the RBC's trajectory approaches the boundary region shown as $v_c$. (**C**) Four measurements were made to establish each subject's baseline before controlled blood loss. Additional measurements were made 1–3 days and 21 days later. (**D**) The modeling integrated serial CBCs into the parameter estimation process in a piecewise manner. The first CBC (left) is assumed to be at steady state, and the model is used to estimate dynamic parameters which produce $RBC_1$ given $RET_1$. These model parameters and $RET_1$ are then used to estimate the initial condition leading to timepoint $t_2$, and the model estimates the dynamics between timepoints $t_1$ and $t_2$. These steps for timepoint $t_2$ are then repeated to estimate the transient dynamics between each successive timepoint. $LS$ refers to the lifespan of RBCs. Panels (**E–F**) are frames from *Video 1* that shows a simulation of the evolution of $P(v,h,t)$ from $t = 0$ to $t = 105$ days for a typical study subject. Equal-probability contours for $P(v,h)$ are shown at the bottom, with the empirical measurement as blue lines, and the simulation in solid red. The surface plot also shows the simulated $P(v,h,t)$. The plot of the empirical measurement in dashed blue is serially updated during the movie to the measurement subsequent to the value of $t$. Marginal $P(v,t)$ and $P(h,t)$ are shown on the left and right.

left (low $u$). Static averages of marginal $v$ and $h$ distributions and other bulk blood characteristics are essential components of modern clinical diagnosis: HGB (hemoglobin concentration per unit volume blood), hematocrit (HCT, volume fraction of RBCs), mean RBC volume (MCV), mean RBC hemoglobin mass (MCH), mean RBC hemoglobin concentration (MCHC), and the coefficient of variation in RBC volume (red cell distribution width or RDW). The ~100,000 single-cell measurements in each routine CBC do not currently directly inform clinical care, but they have great potential to do so.

Anemia (low HGB or HCT) (*Beutler and Waalen, 2006*) is associated with almost all major diseases including cancer, infection, heart failure, autoimmune disease, and malnutrition, and is often the first sign of many of these major illnesses. Understanding the single-cell dynamics of the homeostatic response to blood loss will provide insight into the development and progression of many diseases and enhance our ability to diagnose, monitor, and intervene most effectively.

## Results

### RBC population dynamics can be approximated with a semi-mechanistic unsteady-state mathematical model of RBC volume and hemoglobin and routine CBCs

A routine CBC samples the two-dimensional single-RBC volume-hemoglobin distribution (*P(v,h,t)*) in a patient's circulation at time *t* (*Figure 1*). The composition of the circulating RBC population is determined by dynamic processes: production (erythropoiesis) (*Bunn, 2013*), maturation and aging over a ~ 100-day lifespan (*Willekens et al., 2008*), and clearance (*Franco, 2009*). Master equations are often used to model multi-dimensional probability distributions of single-cell states (*Van Kampen, 2007*). In the case of RBCs, *P(v,h,t)* is determined by a time-dependent production term (*b(v,h,t)*), dynamics, and a clearance term (*d(v,h,t)*). Each routine CBC with a reticulocyte count provides an estimate of both b(v,h,t) and P(v,h,t). The dynamics of P(v,h) can be modeled as a drift-diffusion process ($\nabla(Pf) + \nabla(D\nabla P)$), and the functional specification of the drift, diffusion, and clearance terms can be guided by existing knowledge of in vivo RBC volume and hemoglobin dynamics (*Bosman et al., 2008*; *Franco, 2009*; *Gifford et al., 2006*; *Waugh et al., 1992*; *Willekens et al., 2008*). This overall methodology has also been applied recently to many single-cell gene expression data sets (*Shalek et al., 2013*; *Tusi et al., 2018*) and has several strengths when applied to this clinical data: (1) (*v,h*) space is sampled far more densely than gene expression space, (2) (*b(v,h,t)*) can be directly sampled with each CBC, (3) rich existing physiologic knowledge of the dynamics of (*v,h*) can guide the functional form of $dP/dt$ (*Lew et al., 1995*; *Waugh et al., 1992*; *Higgins and Mahadevan, 2010*), (4) b(v,h,t) and P(v,h,t) can be repeatedly sampled more frequently (minutes) than the characteristic timescale in the system (~100 day RBC lifespan), and (5) inferred single-cell trajectories can easily be combined with electronic medical record data to understand phenotypic effects of dynamics and feedback.

We investigated RBC population dynamics in a cohort of 28 healthy individuals at baseline and following controlled blood loss. We describe the evolution of *P(v,h,t)* with the following equation:

$$\frac{\partial P}{\partial t} = -\nabla \cdot (Pf) + \nabla \cdot (D\nabla P) + b(v,h) - d(v,h) \tag{1}$$

Prior analysis under the assumption of steady state found that the drift term can be approximated as a function of the RBC's current (*v,h*) with an early fast phase of volume and hemoglobin reduction during which the hemoglobin concentration ([Hb]) of young RBCs approaches the population mean (*Higgins and Mahadevan, 2010*). This fast phase is parameterized by $\beta_v$ and $\beta_h$ and is followed by a slower phase of coordinated volume and hemoglobin reduction parameterized by $\alpha$. (See *Figure 1* and details in Materials and methods.) The diffusive term $\begin{bmatrix} D_v & 0 \\ 0 & D_h \end{bmatrix}$ is assumed constant without interaction and encapsulates the variation in the rates of volume and hemoglobin change from one RBC to the next and for the same RBC over time. Based on prior work (*Higgins and Mahadevan, 2010*; *Patel et al., 2015*), the clearance term is approximated as a function of the RBC's current (*v, h*) and a parameter (*$v_c$*) for a clearance boundary region (see *Figure 1* for a schematic).

### The homeostatic response to 10% loss of blood volume includes both an increase in RBC production and a delay in RBC clearance

We studied the effect of blood loss on transient RBC population dynamics by collecting one unit of blood (~10% blood loss) from each subject and estimating model parameters before and after. Significant blood loss triggers a rapid acellular fluid shift to restore intravascular volume that can be detected as a decrease in HCT or HGB. See *Figure 2*. RBCs are assumed to be lost in a volume- and hemoglobin-independent fashion, meaning that *P(v,h,t)* is not directly altered (*Figure 2A*). This

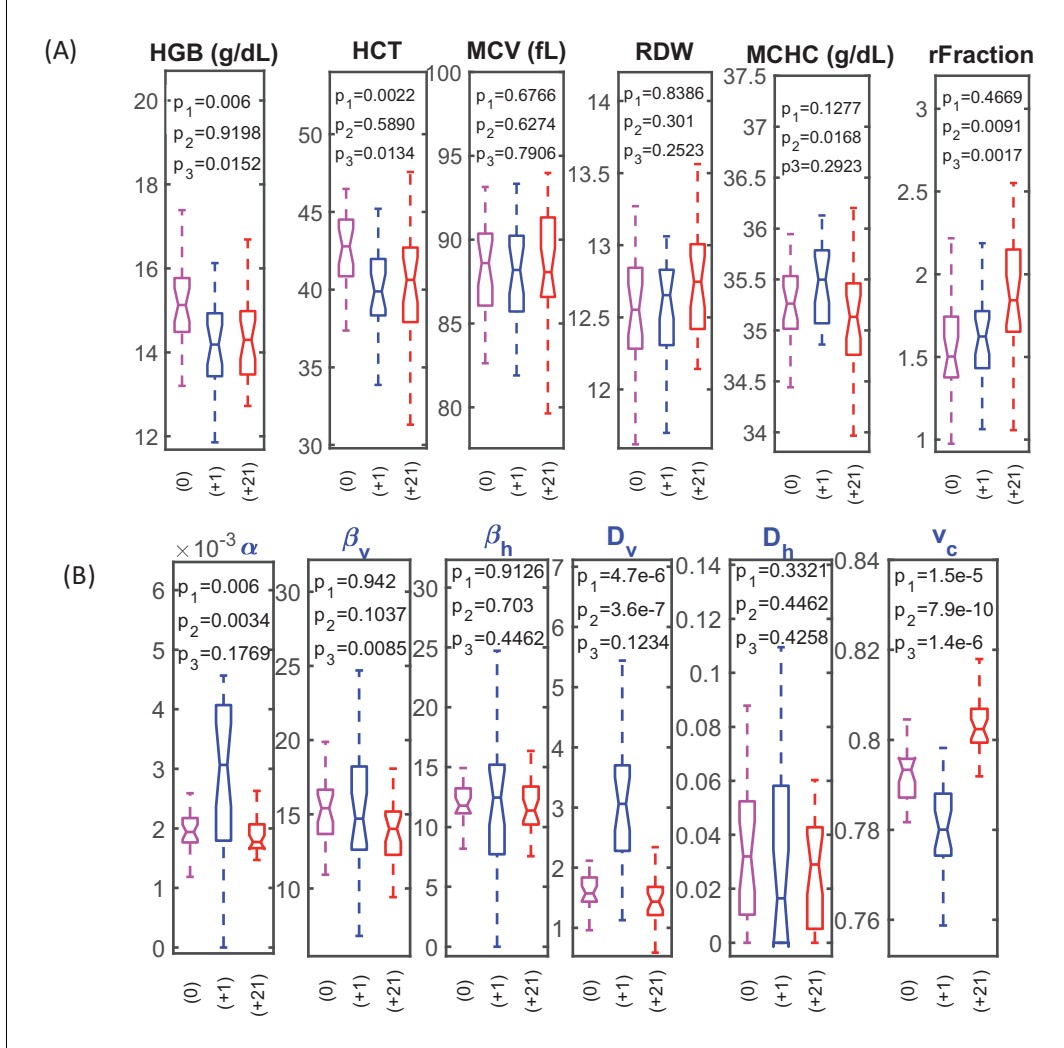

**Figure 2.** RBC dynamics are more sensitive to blood loss than RBC population statistics. (A) Complete blood count (CBC) statistics for 28 healthy subjects before (0), 1–3 days after blood loss (+1), and 21 days after blood loss (+21). Intensive quantities (HGB, concentration of hemoglobin per unit volume of blood; HCT, volume fraction of RBCs in the blood) change significantly immediately following blood loss due to fluid shift, but single-RBC population statistics do not change significantly. MCV, mean RBC volume; RDW, coefficient of variation in RBC volume; MCHC, mean RBC hemoglobin concentration; rFraction, percentage of identified reticulocytes. See *Figure 2—figure supplement 1* for rMCV, mean reticulocyte volume; rRDW, coefficient of variation in reticulocyte volume; MCH, mean RBC hemoglobin mass; CHDW, coefficient of variation in single-RBC hemoglobin concentration. By 21 days after blood loss, the CHDW and rFraction have increased significantly relative to baseline. MCHC at 21 days has decreased relative to 1–3 days. See main text and supplementary information for more detail. (B) Single-RBC volume and hemoglobin dynamics show significant change soon after blood loss. $\alpha$ and $D_v$ increase significantly, and $v_c$ drops. ($p_1$ compares +0 with +1, $p_2$ compares +1 and +21, $p_3$ compares 0 and +21.) Boxplots show the median (middle horizontal line), the 25th and 75th percentiles, and whiskers extend to data points not more than 1.5-times the interquartile range from the median. Notches show a 95% confidence interval for the median, and any additional outliers are shown as discrete points.

The online version of this article includes the following source data and figure supplement(s) for figure 2:

**Source data 1.** Source data for boxplots in *Figure 2*.
**Figure supplement 1.** RBC dynamics are more sensitive to blood loss than RBC population statistics.
**Figure supplement 1—source data 1.** Source data for boxplots in *Figure 2—figure supplement 1*.

assumption is based on prior labeling studies which model the residual lifespan of labeled RBCs

(after reinfusion and recollection) to infer that a blood draw is a random sample of RBCs of all ages (*Franco, 2009*; *Franco et al., 2013*; *Khera et al., 2013*; *Shrestha et al., 2016*). The evidence for this assumption is indirect, relying on models of RBC lifespan distributions, and definitive establishment of its validity awaits the development of an accepted direct measurement or marker of RBC age. An individual can compensate for blood loss by increasing the rate of RBC production or by reducing the rate of clearance, or both. Production and clearance have baseline rates of ~1% per day (*Dornhorst, 1951*; *Franco et al., 2013*). Under physiologic conditions, only the oldest RBCs are cleared (*Cohen et al., 2008*; *Franco, 2009*; *Franco et al., 2013*; *Khera et al., 2013*). The gold standard 'reticulocyte count' does not reliably detect increased production for about 5 days (*Jelkmann and Lundby, 2011*; *Piva et al., 2015*; *Sieff, 2017*) (*Figure 2A*), but the true production rate may increase earlier, and even less is known about any modulation of RBC clearance (*Higgins and Mahadevan, 2010*; *Malka et al., 2014*; *Patel et al., 2015*).

Over the first 1–3 days following blood loss, the single-cell $(v,h)$ dynamics for most subjects showed significant increases in model parameters $\alpha$ and $D_v$ and a decrease in $v_c$ (*Figure 2B*). Greater $\alpha$ reflects a faster reduction in $(v,h)$ for the typical RBC or a longer RBC lifespan, since $\alpha$ is normalized by a nominal lifespan, or both. Greater $D_v$ reflects increased variation in the rate of RBC volume reduction, or a longer RBC lifespan, or both. Smaller $v_c$ reflects delayed clearance of RBCs with $(v,h)$ low enough to have been cleared prior to blood loss.

Model simulation identifies regions of $P(v,h)$ where the blood loss response causes the largest changes (*Figure 3A*): increase in the low-$u$ region containing older cells, milder increase in the high-$u$, low-[Hb] region containing young RBCs, and a balancing decrease along the $u$ axis above the low tail. We can quantify the empirical effect of blood loss response on the older cell fraction by integrating $P(u)$ one standard deviation below the median and lower. *Figure 4D* shows a significant increase in the fraction of older RBCs for most subjects during the first 1–3 days after blood loss, consistent with a delayed clearance.

Newly produced RBCs have higher volume and lower hemoglobin concentration (*d'Onofrio et al., 1995*) and appear in the upper right of the $(v,h)$ plane, or the bottom right quadrant of the $u$-[Hb] plane (*Figure 4AB*). *Figure 3* shows that a simulated increase in $D_v$

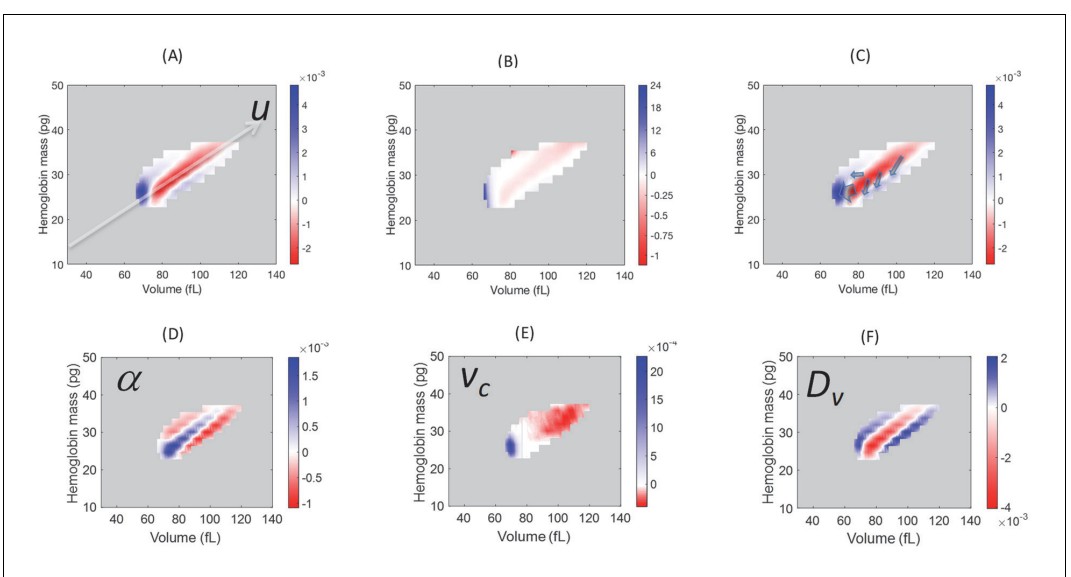

**Figure 3.** Model simulations show that blood loss causes a shift of probability density from the central axis of the $(v,h)$ distribution, mostly to the low volume-low hemoglobin tail. Comparison of the absolute (**A**) and relative (**B**) changes in the simulated single-RBC volume-hemoglobin probability density when setting $D_v'=4D_v$, $\alpha'=2\alpha$, and $v_c'=0.9v_c$, to match the median changes shown in Figure 2B. (**C**) Arrows depict the typical movement in probability density 1–3 days after blood loss. (**D–F**) show the effects of isolated changes to individual parameters, with changes to $\alpha$ and $v_c$ corresponding to retention of older RBCs (delayed clearance), and changes to $D_v$ adding density in the high-volume, low-hemoglobin region where new RBCs appear, corresponding, in part, to increased production.

is associated with an increase in $P(v,h)$ in this region. We can look for empirical evidence of increased production by conditioning on $u$ being more than one standard deviation above the median and then integrating the marginal [Hb] distribution falling at least one standard deviation (~5%) below the median. *Figure 4C* shows a significant increase for the typical subject, consistent with RBC production increasing days earlier than the current gold standard reticulocyte count (*Figure 2A*). We did not find any statistically significant sex-specific differences.

## MCHC rise and subsequent fall is consistent with a combination of delayed clearance and increased production

Single-RBC hemoglobin concentration ([Hb]) increases during the first few weeks of an RBC's lifespan and is then stable (*Franco et al., 2013*). Clearance delay would therefore enrich the fraction of older RBCs which have [Hb] slightly higher than the population mean, and the population mean [Hb] (MCHC) would increase. On the other hand, increased production in isolation would reduce MCHC

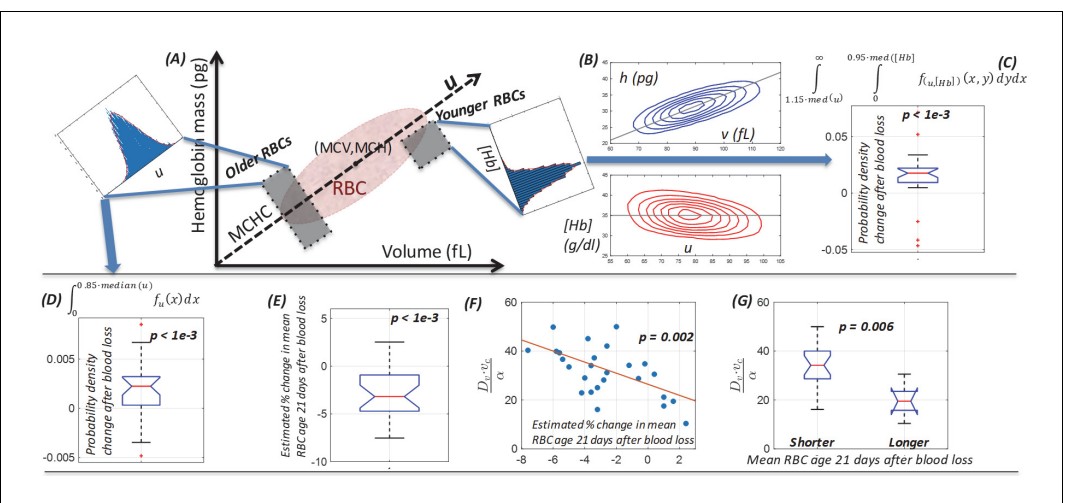

**Figure 4.** Single-cell model provides a mechanistic link between dynamics of the (v,h) distribution and the balance between increased RBC production and delayed RBC clearance in response to blood loss. (A) Schematic of the single-cell volume-hemoglobin distribution for RBCs. The major axis of the distribution (u) corresponds to the mean single-RBC hemoglobin concentration (MCHC). An RBC's position when projected onto u corresponds roughly to its age, with younger RBCs generally appearing in the upper right, and aging along the u axis toward the origin in the bottom left. We can compare changes in the fraction of older RBCs by integrating density along u as shown in the inset in the top left. We can compare changes in the fraction of newly produced RBCs by conditioning on higher u and integrating density along the [Hb] axis as show in the inset in the top right of panel (A). (B) The top panel shows a typical (v,h) distribution that has been transformed onto the u-[Hb] plane in the bottom panel. (C) The typical blood loss response after 1–3 days includes an increase in the fraction of newly produced cells which will have [Hb] more than one standard deviation below the median and u more than one standard deviation above the median (p<1e-3), corresponding to the top right inset in panel (A) and consistent with increased production. (D) 1–3 days following blood loss, the typical response also involves an increase in the fraction of older RBCs, located more than one standard deviation (15%) below the median u (p<1e-3), corresponding to the top left inset in panel (A) and consistent with a delayed clearance. (E) The mean RBC age (M$_{RBC}$), as estimated by the glycated hemoglobin fraction, has decreased on average by about 4% after 21 days, but there is significant variation, with some subjects seeing an increase in M$_{RBC}$. (F) The model characterizes the relative balance between increased production and delayed clearance in each subject's blood response by the dimensionless parameter ratio $(D_v \cdot v_c)/\alpha$. The time-weighted average of this ratio after blood loss for each subject is significantly correlated with the estimated change in M$_{RBC}$ ($\rho = -0.59$), suggesting that the model of (v,h) dynamics has accurately captured the (*production/clearance*) balance of the typical subject's blood loss response. The red line is a least-squares linear fit. (G) The dimensionless parameter ratio distinguishes subjects whose M$_{RBC}$ becomes shorter (production-dominated) during response to blood loss from those whose M$_{RBC}$ becomes longer (clearance-dominated). (See *Figure 2* caption for boxplot description.)

The online version of this article includes the following source data for figure 4:

**Source data 1.** Source data for boxplots in *Figure 4*.

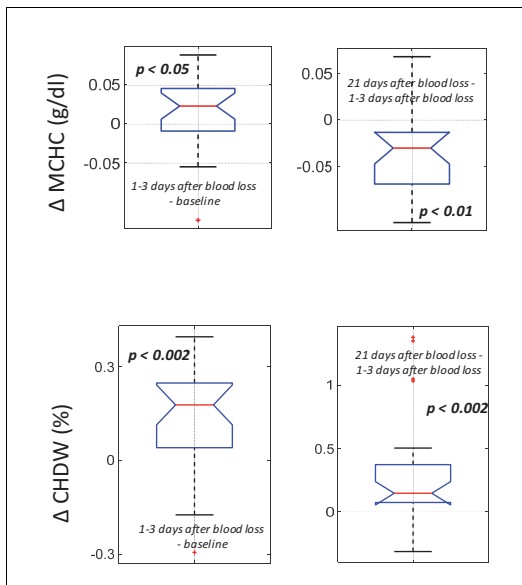

**Figure 5.** Following blood loss, the MCHC rise and fall and the sustained CHDW rise are consistent with a combination of delayed RBC clearance and increased RBC production. (Top) The intra-subject MCHC tends to increase immediately after blood loss (left, p<0.05) and then decreases below baseline by 21 days later (right, p<0.01). (Bottom) The intra-subject CHDW increases immediately after blood loss (p<0.002) and then increases again by 21 days later (p<0.002).

The online version of this article includes the following source data for figure 5:

**Source data 1.** Source data for boxplots in *Figure 5*.

by adding more young RBCs with lower [Hb]. For the typical subject, we find that MCHC increases shortly after blood loss and then falls, dropping below the baseline level by 21 days (*Figure 5*). Both delayed clearance and increased production would be expected to increase the coefficient of variation in [Hb], (cellular hemoglobin distribution width or CHDW) by enriching for RBCs with extreme [Hb], also consistent with measurements of CHDW (*Figure 5*), which increases after blood loss and remains elevated relative to baseline even 21 days later.

## The model enables estimation of the relative magnitudes of the production increase and clearance delay for individual subjects

The model thus suggests that the response to blood loss includes both delayed clearance (modeled as a higher $\alpha$ and lower $v_c$, or simply higher $\frac{\alpha}{v_c}$) and increased production (modeled as a higher $D_v$). These two component responses will have opposite effects on the mean RBC age ($M_{RBC}$), with increased production enriching for younger RBCs and shortening $M_{RBC}$, and delayed clearance enriching for older RBCs and lengthening $M_{RBC}$. $M_{RBC}$ can be estimated in these nondiabetic subjects by measuring the glycated hemoglobin fraction (*Dornhorst, 1951*; *Franco, 2009*; *Khera et al., 2013*; *Malka et al., 2016*; *Bunn et al., 1976*; *Cohen et al., 2008*; *Dijkstra et al., 2017*). *Figure 4* shows that this estimated $M_{RBC}$ has decreased by about ~4% for the typical subject by 21 days, consistent with relatively more increased production than delayed clearance for the typical subject, but the balance varies across subjects.

The model can be used to estimate the $\frac{production}{clearance}$ response ratio for each subject as a dimensionless number: $\frac{D_v v_c}{\alpha}$. Higher $\frac{D_v v_c}{\alpha}$ corresponds to greater production increase and would be expected to shorten $M_{RBC}$, while lower $\frac{D_v v_c}{\alpha}$ corresponds to greater clearance delay and would lengthen $M_{RBC}$. We validate the model by comparing $\frac{D_v v_c}{\alpha}$ to the change in $M_{RBC}$ estimated from independent measurements of HbA1c and find (*Figure 4F*) the expected negative correlation ($p < 0.002$). Subjects whose modeled blood loss response shows transient ($v,h$) dynamics with relatively higher production increase have a greater reduction in $M_{RBC}$ (*Figure 4G*).

## Perturbations to single-RBC volume and hemoglobin distributions persist for at least 21 days after loss

The model thus finds that volume and hemoglobin dynamics of the typical RBC are significantly altered shortly after blood loss and remain altered for at least 21 days. Because $P(v,h,t)$ is determined by these dynamics, our results imply that it should be possible to distinguish 21-day post-blood loss CBCs from pre-blood loss CBCs based only on $P(v,h)$, without having to consider measurements of cell count or concentration like HGB, HCT, or reticulocyte count. We used machine learning methods to classify measurements of $P(v,h)$ and achieved cross-validated performance > 98% (AUC 0.98) with multiple methods (quadratic discriminants, complex trees, etc.). By comparison, this classification by P(v,h) was actually significantly more accurate than classification using only the currently standard count-based markers (HCT and reticulocyte count, accuracy 93%, AUC 0.90).

## Discussion

This single-cell model of routinely available clinical data provides a mechanistic link between the (*v*, *h*) distribution and changes in the RBC age distribution. The model identifies delayed RBC clearance as an important unrecognized component of the compensatory response to blood loss, and it enables more nuanced and precise inferences about the homeostatic response to a fundamental pathologic process in different individuals.

Our analysis begins with a mechanistic model and leads to identification of empirical changes in the (*v*,*h*) distribution that are associated with the response to blood loss. A non-mechanistic approach comparing arbitrary distribution statistics before and after blood loss may also be fruitful, but given the large number of potential statistics on distributions of tens of thousands of measurements and the small number of cases (n = 28), statistical significance of the identified associations would likely be limited. More importantly, the advantage of a mechanistic modeling approach either in addition to or instead of a purely statistical or machine learning approach is that it provides a hypothesized physiologic context. Additional falsifiable predictions may then be deduced to provide further validation opportunities, as shown for instance in *Figure 5*. A mechanistic model also enables assessment of counterfactuals, which is particularly important in the clinical context, where patient factors or pre-existing conditions not present in discovery or development cohorts might significantly compromise accuracy when inference methods are applied to real-world populations. An understanding of the mechanistic basis for an inference method or algorithm will increase the likelihood that these problematic situations can be anticipated and perhaps avoided. In the context of this study, such conditions may include transfusion, sickle cell disease, or mechanical RBC stresses altering RBC volume associated with disseminated intravascular coagulation, microangiopathic hemolytic anemia, and other related pathologic processes.

The model has potential for immediate clinical decision support by detecting increased RBC production earlier than the current gold standard reticulocyte count in our study cohort. Further study is needed to compare the transient (*v*,*h*) dynamics in patients with active disease processes and to investigate which factors control the production/clearance ratio of a subject's blood loss response. As more single-cell methods mature, modeling of higher-dimensional cell states will enable richer understanding of physiologic homeostasis and adaptation and help realize the vision for precision medicine.

## Materials and methods

### Human subjects

All 28 subjects (18 male, 10 female) enrolled in the study were healthy and athletically active individuals aged 18 to 40 on the day of enrollment. The study size provided at least four same-sex biological replicates and allowed for the possibility of a 50% dropout during the study. Subjects were excluded from enrollment if they participated in competitive sporting events during the study procedures, or if they were a member of a registered anti-doping testing pool for any international sporting federations, national anti-doping organizations, or professional sporting organizations. Prior to study commencement, all participants provided written, informed consent. Approval for study procedures was granted by the University of Utah Institutional Review Board (IRB Protocol #00083533) and for analysis of human subject data by the Partners Healthcare Institutional Review Board. An outline of the study design and collection time points is shown in *Figure 1*.

### Blood collection

Prior to each blood collection, subjects were seated with their feet on the floor for a minimum of ten minutes per World Anti-Doping Agency blood collection guidelines (https://www.wada-ama.org/en/resources/world-anti-doping-program/guidelines-blood-sample-collection). After the ten-minute equilibration period, blood was collected via venipuncture of an antecubital vein into one 6 mL serum-separator tube and one 6 mL $K_2$EDTA (BD Vacutainer) tube. After collection, whole blood samples were immediately refrigerated until analysis. Additional aliquots were stored at $-80C$ for HbA1c measurement. Following three baseline collections over the course of 2-4 weeks, each subject in the study donated one unit of blood (~475 mL) according to Associated Regional and University Pathologists (ARUP) standard operating procedures.

## CBC measurements

Whole blood samples collected in K$_2$EDTA tubes were measured for a Complete Blood Count plus reticulocyte% using a Siemens Advia 2120i. Briefly, samples were brought from refrigerated to room temperature while on a nutating mixer for at least 15 min prior to analysis. All samples were measured in duplicate. All samples were collected in Salt Lake City, Utah, at either the Sports Medicine Research and Testing Laboratory (SMRTL) or the University of Utah Hospital. The approximate altitude at these locations is 1400 m above sea level. All subjects in the study were residents at this altitude and are assumed to be adapted to the environment.

## Model details

We measured CBCs roughly every other day for a week for all subjects and used the model to infer each subject's baseline RBC population dynamics between these 4 timepoints (*e.g.*, *t = 1, 3, 5,* and *7 days*). At t = 1, *b(v,h,1)* is measured and used to estimate source terms extending back in time by a number of days equivalent to the RBC lifespan (*LS*): $b(v, h, (1 - LS) <= t < 1) = b(v, h, t = 1)$. The RBC age distribution is assumed to be uniform with nominal $LS = 105 \ days$ (**Cohen et al., 2008**). The first CBC provides a sample of *P(v,h,1)*, and *Equation 1* can be used to estimate the parameters characterizing the RBC population dynamics at baseline: $p_1 = (\alpha_1, \ \beta_{v,1}, \beta_{h,1}, D_{v,1}, D_{h,1}, v_{c,1})$ (**Higgins and Mahadevan, 2010**; **Patel et al., 2015**). The transient dynamics between *t = 1* and *t = 3* can be estimated using $p_1$ and *Equation 1*. Initial conditions at *t = 1* are determined by integrating *Equation 1* for *LS – 2* days with a source term equal to *b(v,h,1)*. The CBC measured on day 3 (*t = 3*) provides a direct estimate of *b(v,h,3)* and a sample of *P(v,h,3)*. *Equation 1* is then used to estimate $p_3$, the parameters characterizing the transient dynamics between *t = 1* and *t = 3*. This process is repeated for each successive CBC to provide quantification of the transient dynamics as shown in *Figure 2*. See *Video 1* for additional detail.

In the Fokker-Planck equation describing the RBC maturation dynamics (*Equation 1*), the drift term is expressed as a combination of an initial fast phase, followed by a slow phase. In *Equation 2*, *v* and *h* are normalized by their sample population means, and both approach 1 as the fast phase transitions to the slow phase:

$$f = \frac{\alpha e^{\beta_v(v-h)}}{\alpha e^{\beta_h(h-v)}} \tag{2}$$

In *Equation 1 and 2*, *P* refers to the volume-hemoglobin probability distribution of the RBC population, *D* is the diffusion matrix $\begin{bmatrix} D_v & 0 \\ 0 & D_h \end{bmatrix}$, and $\alpha$, $\beta_v$, and $\beta_h$ parameterize the drift processes. The birth term *b(v,h,t)* is estimated by reticulocyte count measurements at time *t* along with the RBC population, with *b(v,h)* defined by the volume and hemoglobin distribution measured for reticulocytes identified using standard validated clinical laboratory techniques (**d'Onofrio et al., 1995**). The clearance term, *d(v,h)* is defined as follows:

$$d(v, h) = \frac{1}{1 + e^{\Delta(v,h)}}$$

$$\Delta(v, h) = 100 \frac{\cos(\theta)\sqrt{(v\bar{v})^2 + (h\bar{h})^2} - v_c\sqrt{\bar{v}^2 + \bar{h}^2}}{v_c\sqrt{\bar{v}^2 + \bar{h}^2}}$$

$$\theta = tan^{-1}\left(\frac{\bar{h}}{\bar{v}}\right) - tan^{-1}\left(\frac{h\bar{h}}{v\bar{v}}\right)$$

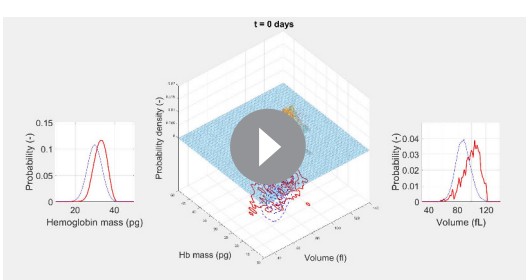

**Video 1.** The video shows a simulation of the evolution of *P(v,h,t)* from *t = 0* to *t = 105 days* for a typical study subject. Equal-probability contours for *P(v,h)* are shown at the bottom, with the empirical measurement as blue dashed lines, and the simulation in solid red. The surface plot also shows the simulated *P(v,h,t)*. The plot of the empirical measurement in dashed blue is serially updated during the movie to the measurement subsequent to the value of *t*. Marginal distributions, *P(v,t)*, and *P(h,t)*, are shown at the sides along with empirical measurements in blue.
https://elifesciences.org/articles/48590#video1

Here, $\bar{v}$ and $\bar{h}$ are the MCV and MCH, respectively, and $v_c$ parameterizes the clearance boundary region (*Higgins and Mahadevan, 2010*; *Patel et al., 2015*).

## Acknowledgements

This study was funded by the NIH, the Life Sciences Research Foundation (LSRF), and the Partnership for Clean Competition. None of the funders played any role in the decision to submit for publication. The authors appreciate expert advice on HbA1c testing from Dr. Randie Little and the technical assistance of the Diabetes Diagnostic Laboratory at the University of Missouri Medical School. The authors acknowledge instrument and testing support from Siemens Healthcare Diagnostics and Sebia Diagnostics. AC is a Good Ventures Fellow of the Life Sciences Research Foundation. All simulations were run on the Harvard Medical School O2 cluster. The authors would like to thank Jonathan Carlson, Bronner Goncalves, Michael Dworkin, Erica Normandin, Charles Pedlar, and Rebecca Ward for helpful discussions.

## Additional information

### Funding

| Funder | Grant reference number | Author |
| --- | --- | --- |
| National Institutes of Health | 1DP2DK098087 | John M Higgins |
| Partnership for Clean Competition | | Daniel Eichner John M Higgins |
| Life Sciences Research Foundation | Good Ventures Fellowship | Anwesha Chaudhury |

The funders had no role in study design, data collection and interpretation, or the decision to submit the work for publication.

### Author contributions

Anwesha Chaudhury, Conceptualization, Data curation, Software, Formal analysis, Validation, Investigation, Visualization, Methodology, Writing—original draft, Project administration, Writing—review and editing; Geoff D Miller, Conceptualization, Resources, Data curation, Investigation, Methodology, Project administration, Writing—review and editing; Daniel Eichner, Conceptualization, Resources, Data curation, Funding acquisition, Investigation, Methodology, Project administration, Writing—review and editing; John M Higgins, Conceptualization, Resources, Data curation, Software, Formal analysis, Supervision, Funding acquisition, Validation, Investigation, Visualization, Methodology, Writing—original draft, Project administration, Writing—review and editing

### Author ORCIDs

Anwesha Chaudhury https://orcid.org/0000-0002-9945-6862
John M Higgins https://orcid.org/0000-0002-9182-0076

### Ethics

Human subjects: Approval for study procedures was granted by the University of Utah Institutional Review Board (IRB Protocol #00083533) and for analysis of human subject data by the Partners Healthcare Institutional Review Board.

### Decision letter and Author response

Decision letter https://doi.org/10.7554/eLife.48590.sa1
Author response https://doi.org/10.7554/eLife.48590.sa2

## Additional files

### Supplementary files
• Transparent reporting form

### Data availability
All data we are authorized to share according to our Institutional Review Board approved study protocols for patient data collection and subsequent analysis and publication is included in the manuscript. We are able to share de-identified study subject data, and we have provided source data files for all appropriate figures as tables in spreadsheets.

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
