## [Decision Letter]

**Acceptance summary:**

The paper uses experimentally induced blood loss to study red cell dynamics, with the innovation of using single-cell measurements before and after blood loss to parameterize a model of red cell birth, dynamics, and clearance. This allows the authors to draw conclusions about production of red blood cells (detectable earlier than current methods) and previously unsuspected decreases in clearance. The paper is a good example of how the kinds of rich data routinely collected by clinical labs, but often ignored in clinical practice, can be made sense of in the service of both clinical care – here, early detection of blood loss – and physiological understanding of biological processes.

**Decision letter after peer review:**

Thank you for submitting your article "Single-Cell Modeling of Routine Clinical Blood Tests Reveals Transient Dynamics of Human Response to Blood Loss" for consideration by *eLife*. Your article has been reviewed by two peer reviewers, and the evaluation has been overseen by Naama Barkai as the Senior and Reviewing Editor. The following individuals involved in review of your submission have agreed to reveal their identity: Ziad Obermeyer (Reviewer #1); Steven Spitalnik (Reviewer #2).

The reviewers have discussed the reviews with one another and the Reviewing Editor has drafted this decision to help you prepare a revised submission.

As you can see, both reviewers appreciated the importance of your analysis and findings, and supported publication. Provided that the concerns of reviewer #1 can be satisfactory addressed, the paper can be accepted.

Reviewer #1:

This paper uses experimentally induced blood loss to study red cell dynamics. The innovation is to use single-cell measurements from this setup, before and after blood loss, to estimate parameters of a model of red cell birth, dynamics, and clearance. This allows the authors to draw conclusions about production of RBCs (detectable earlier than current methods) and previously unsuspected decreases in clearance.

Overall I found the paper to be a compelling early example of how the kinds of rich data routinely collected by clinical labs, but generally ignored in clinical practice, can be made sense of in the service of both clinical care (e.g. early detection of blood loss) and physiological understanding.

I had a few questions about the assumptions – to the extent these concerns are unfounded, I would have appreciated more explanation of them in the Materials and methods.

- The authors state that blood loss is essentially a random draw of RBCs (i.e. that the distribution P is unchanged before and after). Is this assumed or known? I can imagine several scenarios where different kinds of cells could be more or less likely to be in capillaries vs. larger vessels accessed by venipuncture. I can also imagine that such non-random changes induced directly by drawing blood would pose problems for estimating model parameters as the authors do, and I would have appreciated reassurance or some discussion of this.

- The birth function is estimated by measuring reticulocytes, but I could not tell whether reticulocytes were inferred from the size distribution (i.e. by assuming they are drawn from some distribution of high (v,h)) or measured directly. If the former, how sensitive are measurements to the assumptions used to define reticulocytes?

- What altitude were the blood draws taken at? And how does the sample frame how we should interpret results? I ask because patients seem to be athletes in Utah – things might be different in Park City vs. Boston.

I have a broader point on framing. One of the compelling practical aspects of the paper is the idea that the model parameters can be used as a new way to infer early blood loss.

- The authors set up the current CBC parameters as the straw man, and note that their methods perform better in a variety of ways. Fair enough. But these former measures are almost laughably simple – two means and two variances (even the second variance is, if I'm not mistaken, not commonly reported). A better comparison would be some more sophisticated measures of the marginal v and h distributions, and especially measures of covariance.

- It's possible that a larger set of X's (right hand side variables) such as these would do just as well as the model derived parameters in predicting whether a measurement was taken before or after blood loss, particularly if fed into a good machine learning model – after all, the model is picking up on some empirical shifts in distribution and (given infinite data) it would be impossible for the structural model to do better at predicting something than a good prediction model itself. If this is not true, all the better – I can easily imagine that in small samples the model does much better than a kitchen sink + ML approach, but this is in itself worth showing.

- Regardless, one of the primary benefits of having a structural model of the physiology (as opposed to a bunch of measurements + ML) seems to be to perform counterfactual simulations. While this is not my area, I can imagine a number of interesting questions – how would the dynamics change under a range of different conditions: different volumes or chronicity of blood loss (e.g. from colon cancer rather than a unit of drawn blood), etc. One could also specifically model the kinds of changes that would be detected by single cell measures but specifically not by standard measures.

Finally, as a style point, I was a bit overwhelmed by all the figures. Many of the subfigures were not even discussed in the text, which may be a sign that they belong as supplementary Figures. I found the video quite informative (though would have liked the marginals projected as well) and wonder if putting a few frames from this as a figure would help give intuitions about what is actually happening empirically. Overall, refocusing on the main innovations of the paper and cutting some unnecessary material would be helpful to the reader.

Reviewer #2:

This is the latest in a series of interesting and provocative studies from Dr. Higgins and his colleagues. They have identified novel ways of "mining" data from routine CBCs to provide additional clinical insights and identify underlying mechanisms and/or opportunities for further research. This manuscript similarly succeeds in these regards.

In particular, by studying otherwise healthy volunteers, they identify that some individuals respond to an acute blood loss by, predominantly, rapidly producing new RBCs, whereas others respond by, predominantly, slowing down clearance of existing, circulating RBCs. To my knowledge, these are new and very interesting findings, particularly the latter. What distinguishes these individuals in their predominant response characteristics? Genetics? Diet? Environmental influences? Other things? This will provide a rich opportunity for future studies.

In addition, it will be interesting, in the future, to investigate how various patient populations, with various underlying disorders, respond to acute blood loss, whether that blood loss is pathological (e.g., a GI bleed or trauma) or iatrogenic (e.g., during and following surgery). Unravelling the underlying mechanisms will be important in expanding our knowledge of basic pathophysiology and may also affect how physicians respond therapeutically.

Finally, although one can provide plausible underlying mechanisms, based on prior work, regarding how humans respond to acute blood loss by increasing RBC production, it is harder to conceive of how clearance of aging RBCs is regulated in this setting. How does the mononuclear phagocyte system "recognize" acute blood loss and then down-regulate clearance accordingly? This interesting conundrum opens an important new area for investigation.

---

## [Author Response]

Reviewer #1:[…] I had a few questions about the assumptions – to the extent these concerns are unfounded, I would have appreciated more explanation of them in the Materials and methods.

The reviewer raises valid questions about assumptions and our approach that we address in our revised manuscript as suggested, with details below.

- The authors state that blood loss is essentially a random draw of RBCs (i.e. that the distribution P is unchanged before and after). Is this assumed or known? I can imagine several scenarios where different kinds of cells could be more or less likely to be in capillaries vs. larger vessels accessed by venipuncture. I can also imagine that such non-random changes induced directly by drawing blood would pose problems for estimating model parameters as the authors do, and I would have appreciated reassurance or some discussion of this.

We have clarified in our revised Results section that this assumption is based on prior labeling studies which model the residual lifespan distribution of labeled RBCs (after reinfusion and recollection) to infer that a blood draw is a “random sample” consisting of “a mixture of RBCs of all ages.” (Shrestha et al., 2016) We make two further notes. First, the evidence for this assumption is indirect, relying on models of RBC lifespan distributions, because there is no accepted direct measurement or marker of RBC age, and some uncertainty regarding this assumption is warranted, as we now state in our revised manuscript. Second, we agree with the reviewer that factors such as vascular geometry likely do have differential effects on RBCs with different properties, such as volume, and to the extent that these different properties are associated with age, blood draws from these different locations would be expected to provide incompletely random samples of the age distribution. We expect that the magnitude of these differences is less than the analytic precision of the current volume and hemoglobin distribution measurements. We are not aware of any evidence that blood draws from different anatomic sites yield significantly different RBC volume or hemoglobin distributions (though factors like the postural state of the patient do effect HCT and HGB (Leppänen and Gräsbeck, 2009)), but this question has not been systematically investigated as far as we know.

- The birth function is estimated by measuring reticulocytes, but I could not tell whether reticulocytes were inferred from the size distribution (i.e. by assuming they are drawn from some distribution of high (v,h)) or measured directly. If the former, how sensitive are measurements to the assumptions used to define reticulocytes?

We have revised our description of the birth function in our Materials and methods section to make clearer that we are defining it using the standard validated reticulocyte measurement (d’Onofrio et al., 1995), which provides the distribution of volume and hemoglobin masses of reticulocytes.

- What altitude were the blood draws taken at? And how does the sample frame how we should interpret results? I ask because patients seem to be athletes in Utah – things might be different in Park City vs. Boston.

As we now state in our revised Materials and methods, all samples were collected in Salt Lake City, Utah, at approximately 1400 m above sea level, and because all study subjects were residents of the area and were adapted to this altitude, we do not expect altitude to have a significant effect on our results.

I have a broader point on framing. One of the compelling practical aspects of the paper is the idea that the model parameters can be used as a new way to infer early blood loss.- The authors set up the current CBC parameters as the straw man, and note that their methods perform better in a variety of ways. Fair enough. But these former measures are almost laughably simple – two means and two variances (even the second variance is, if I'm not mistaken, not commonly reported). A better comparison would be some more sophisticated measures of the marginal v and h distributions, and especially measures of covariance.- It's possible that a larger set of X's (right hand side variables) such as these would do just as well as the model derived parameters in predicting whether a measurement was taken before or after blood loss, particularly if fed into a good machine learning model – after all, the model is picking up on some empirical shifts in distribution and (given infinite data) it would be impossible for the structural model to do better at predicting something than a good prediction model itself. If this is not true, all the better – I can easily imagine that in small samples the model does much better than a kitchen sink + ML approach, but this is in itself worth showing.

The reviewer raises an important and timely topic. As we now state in our revised Discussion, while our analysis begins with a mechanistic model and leads to identification of empirical changes in the (v,h) distribution that are associated with the response to blood loss, a brute force machine learning approach comparing arbitrary distribution statistics before and after blood loss would also be fruitful. The statistics to which the model leads us are associated with blood loss and would be identified by the machine learning approach as the reviewer notes, as would a very large number of other correlated statistics. The challenge as the reviewer notes is that in this case there is a vastly larger number of potential statistics on distributions of tens of thousands of measurements, and coupled with the small number of cases (n = 28), statistical significance of identified associations would be more difficult if not impossible to establish, if mechanistic insights were not used to bias the prioritization of statistics to investigate.

- Regardless, one of the primary benefits of having a structural model of the physiology (as opposed to a bunch of measurements + ML) seems to be to perform counterfactual simulations. While this is not my area, I can imagine a number of interesting questions – how would the dynamics change under a range of different conditions: different volumes or chronicity of blood loss (e.g. from colon cancer rather than a unit of drawn blood), etc. One could also specifically model the kinds of changes that would be detected by single cell measures but specifically not by standard measures.

As we note in our revised Discussion, we strongly agree with the reviewer that the advantage of the mechanistic modeling approach either in addition to or instead of machine learning is that it provides a hypothesized physiologic context enabling further validation by testing many additional falsifiable predictions derived by logical extension, some of which are mentioned by the reviewer, as well as counterfactuals. In the clinical context, we agree with the reviewer that the ability to assess counterfactuals is particularly important, as is also highlighted by comments from reviewer #2 below: if we understand the mechanistic basis for a clinical predictor, then we will be in a better position to identify underlying disorders that may make the predictor misleading (e.g., transfusion, sickle cell disease, microangiopathic hemolytic anemia, xerocytosis, some medications, etc.).

Finally, as a style point, I was a bit overwhelmed by all the figures. Many of the subfigures were not even discussed in the text, which may be a sign that they belong as supplementary figures. I found the video quite informative (though would have liked the marginals projected as well) and wonder if putting a few frames from this as a figure would help give intuitions about what is actually happening empirically. Overall, refocusing on the main innovations of the paper and cutting some unnecessary material would be helpful to the reader.

We appreciate the good suggestions and have moved the unreferenced figure panels into supplementary figures. We have also added the marginal distributions to the video and have added frames to Figure 1.

Reviewer #2:This is the latest in a series of interesting and provocative studies from Dr. Higgins and his colleagues. They have identified novel ways of "mining" data from routine CBCs to provide additional clinical insights and identify underlying mechanisms and/or opportunities for further research. This manuscript similarly succeeds in these regards.In particular, by studying otherwise healthy volunteers, they identify that some individuals respond to an acute blood loss by, predominantly, rapidly producing new RBCs, whereas others respond by, predominantly, slowing down clearance of existing, circulating RBCs. To my knowledge, these are new and very interesting findings, particularly the latter. What distinguishes these individuals in their predominant response characteristics? Genetics? Diet? Environmental influences? Other things? This will provide a rich opportunity for future studies.

We are very pleased that the reviewer finds our study interesting and provocative. We agree that it will be very interesting to investigate the factors that determine the different individual level responses, and we anticipate that discovery of relevant factors will further elucidate fundamental mechanisms of RBC pathophysiology. Based in part on some of the reviewer’s prior work, we suspect that small changes in iron levels, even within the reference interval, may be relevant, and that is a high-priority area for future investigation but will require a larger study or a design more focused on that hypothesis.

In addition, it will be interesting, in the future, to investigate how various patient populations, with various underlying disorders, respond to acute blood loss, whether that blood loss is pathological (e.g., a GI bleed or trauma) or iatrogenic (e.g., during and following surgery). Unravelling the underlying mechanisms will be important in expanding our knowledge of basic pathophysiology and may also affect how physicians respond therapeutically.Finally, although one can provide plausible underlying mechanisms, based on prior work, regarding how humans respond to acute blood loss by increasing RBC production, it is harder to conceive of how clearance of aging RBCs is regulated in this setting. How does the mononuclear phagocyte system "recognize" acute blood loss and then down-regulate clearance accordingly? This interesting conundrum opens an important new area for investigation.

We agree that evidence for mechanisms modulating RBC clearance is lacking. We speculate that signals enhancing RBC production, such as erythropoietin, might play a dual role and potentially increase the likelihood that a mononuclear phagocyte would down-regulate its clearance activity, but any evidence supporting this hypothesis is currently absent.